Reproductive success of Whiskered Tern Chlidonias hybrida in eastern Spain in relation to water level variation

Ortiz Lledó Álvaro milicora@yahoo.es
Vidal Mateo Javier
Urios Moliner Vicente
Department of Environmental Sciences and Natural Resources, Universidad de Alicante , Alicante , Spain
Griffith Simon
Electronic publication date: 2018 Apr 19
Publication date: 2018
Volume: 6
Electronic Location ID: e4548
Received 2015 Nov 30; Accepted 2018 Mar 8
Copyright: ©2018 Ortiz Lledó et al.
Copyright year: 2018
Copyright holder: Ortiz Lledó et al.
License: This is an open access article distributed under the terms of the Creative Commons Attribution License, which permits unrestricted use, distribution, reproduction and adaptation in any medium and for any purpose provided that it is properly attributed. For attribution, the original author(s), title, publication source (PeerJ) and either DOI or URL of the article must be cited.
License URL: https://creativecommons.org/licenses/by/4.0/

Keywords: Migratory species, Reproductive success, Wetland, Water level fluctuation, Chlidonias hybrida, Whiskered Tern, Hatching success, Fledgling success

Funding: Alicante University, Department of Environmental Sciences and Natural Resources This work was supported by Alicante University, Department of Environmental Sciences and Natural Resources. The funders had no role in study design, data collection and analysis, decision to publish, or preparation of the manuscript.

==============================
Background

A study on the Whiskered Tern Chlidonias hybrida was carried out between 2002 and 2009 in wetlands of eastern Spain to evaluate how water level fluctuation affects its reproductive success (hatching, fledgling and breeding success). This species is catalogued as Vulnerable in Spain and has an unfavorable conservation status in Europe.

Methods

Our study includes 18 sampling areas from five wetlands, covering a total of 663 nests, 1,618 eggs, 777 nestlings and 225 fledglings. The colonies were visited at least twice per week in breeding period. The number of eggs and/or nestlings present in each nest were annotated each time the colonies were visited with the aim to compare the evolution of these parameters with time. Hatching success was calculated as the proportion of egg that hatched successfully. Fledgling success and breeding success were calculated as the proportion of chicks that fledged successfully and the proportion of eggs that produced fledglings. We used the Kruskal–Wallis test to analyze the differences in the dependent variables hatching, fledgling and breeding success among the wetlands and the sampling areas. We explored the relationship between the different reproductive success with the average fluctuation rate and the anchoring depth of nests, using statistics of the linear regression.

Results

It was observed that the reproductive success varied significantly in the interaction among the different categories of water level fluctuation and the different areas (using the Kruskal–Wallis test). Our records showed that pronounced variations in water level destroyed several nests, which affected the Whiskered Tern reproductive success. Considering all events that occurred in 18 areas, the mean (±SD) of nests, eggs and nestlings that were lost after water level fluctuations were of 25.60 ± 21.79%, 32.06 ± 27.58% and 31.91 ± 21.28% respectively, also including the effects of rain and predation.

Discussion

Unfavorable climatic events, such as strong wind, rain or hail, also caused the loss of nests, eggs and nestlings, even when wetland water levels remained constant. The influence of the anchorage depth of the nest and the water level fluctuation rate were analyzed and did not provide statistically significant results. It was not possible to establish a clear pattern on these latter variables, so further studies are needed to obtain more significant results. We propose to undertake similar studies in wetlands where the water level can be regulated, with the range of nest anchorage depth on the emergent vegetation being between 30 and 60 cm, which could improve the reproductive success in this kind of habitats. As recommendation, in water level controlled wetlands (that use sluices), it should not vary more than ±6 cm in a short time (1–2 days) once the nests are established since it negatively affects their reproductive success.

Introduction

The population of Whiskered Tern Chlidonias hybrida (Pallas, 1811), a migratory species of the family Sternidae, has declined in the Iberian Peninsula and Europe from 1970 to 1990 (Urios et al., 1991; Tucker & Heath, 1994). It is catalogued as an Endangered Species in Spain, according to Real Decreto 139/2011 (BOE, 2011), and has an unfavorable conservation status in Europe (Tucker & Heath, 1994), as it is included in the priority conservation of the wild birds directive (BirdLife, 2004).

Between 1990 and 2000, their populations remained relatively stable in Europe, with clear increases in the central and eastern parts of the continent, comprising ca. 87,000 pairs (BirdLife, 2004). In 2015, their populations have been estimated between 66,300 and 108,000 pairs in Europe and ca. 6,400 pairs in Spain (BirdLife, 2015). The future of the species in the Iberian Peninsula appears to be uncertain due to habitat destruction (Callaghan & Villaplana, 1990), urban pressure and the reduction of wetlands due to their transformation in agricultural areas (Villaplana, 1984; Callaghan & Villaplana, 1990; Urios et al., 1993), as well as the increase in pollution and loss of water quality (Blanco & González, 1992), the inadequate management of water reservoirs and lagoons, and climatological factors (Máñez et al., 2004). In the natural wetland of Albufera of Valencia (39°20′05″N, 0°21′08″W, SE Spain) that includes 21,120 ha, about 2,000 pairs of Whiskered Terns nested in 1970. Since then, an important decline of nesting pairs occurred, and in 1980 no more nesting pairs were observed in that area (Urios et al., 1991). Since 1980 to the present the species did not nest in the Albufera of Valencia, although it has been observed resting in this area while migrating.

The reproductive success of the Whiskered Tern greatly depends on numerous factors such as the chosen area for reproduction by parents, the reproduction in colonies or as isolated nests, the climatic conditions, the water level variation in the wetlands and its stability during the reproduction period, the abundance of food (Tucker & Heath, 1994; Catry, Tomé & Cardoso, 1997; Carpentier, Paillisson & Marion, 2002); the type, form and size of the nests, the presence of parasites and predators, the type of vegetation on which the nests are anchored and its abundance (Spina, 1982; Dostine & Morton, 1989; Bakaria et al., 2002; Lautrabe, 2006; Paillisson et al., 2006; Bakaria et al., 2009; Ledwoń et al., 2014); and the quality of cares that parents provide to their descendence (Nichols, Spendelow & Hines, 1990; Spendelow et al., 1995; Brunton, 1997; Lebreton et al., 2003; Paillisson et al., 2007; Álvarez & Barba, 2008; Monticelli et al., 2008; Braasch, Schauroth & Becker, 2009; Ledwoń, Neubauer & Betleja, 2013). When all these conditions are suitable, the reproductive success is usually high.

As a fragile medium disturbance species, adverse conditions in the breeding season force the populations to abandon the colony (Spina, 1982; Catry, Tomé & Cardoso, 1997; Bakaria et al., 2002; Paillisson et al., 2006; Ledwoń, 2011), which cause the loss of nests, eggs and nestlings; force second clutches (Catry, Tomé & Cardoso, 1997; Ortiz, 2005); or cause migration to other areas with more sheltered habitats, and enough available food and water for nesting (Paillisson et al., 2007). This has been observed also in other species of the same family (Buckley & Buckley, 1972; Nisbet, Spendelow & Hatfield, 1995; Stienen & Brenninkmeijer, 2002; Van de Pol et al., 2010).

The main aim of our study was to study the reproductive success of the Whiskered Tern in relation to water level variability. We consider the hypothesis that a strong variation of water level in a short period of time could negatively affect breeding success. Nests of this species are anchored to emergent vegetation (such as: Phagmites sp., Thypha sp, Scirpus sp., Myriophyllum sp.), and therefore changes in the water level in the wetlands where they breed could negatively affect to its reproductive success, since eggs or nestlings could be lost in that process (Van de Pol et al., 2010). This also occurs in other species of the Sternidae family due to natural hydrological regimes that changes the shape of the colonies, their distribution and the number of individuals in a wetland (Atamas & Tomchenko, 2015).

We expect that our study will bring new knowledge to enhance the management and conservation of the species in the wetlands where they nest which could serve as a model for management in other areas, and we discuss recommendations for the conservation of Whiskered Terns.

Study Area & Methods

Study area

The present study was carried out during eight breeding seasons (2002–2009) of Whiskered Tern in the Pego-Oliva Natural Park (Valencian Community, Spain). Additional observations were made at four further wetlands in Valencian Community: Hondo de Elche-Crevillente N.P. (Alicante), Xeresa marsh (Valencia), Moro marsh (Valencia) and Almenara marsh (Castellón). These latter wetlands are located between 20 and 100 km from Pego-Oliva Natural Park in straight line (Fig. 1). The Valencian Community government and managers of the Natural Parks involved in this study provided permissions to work in the cited wetlands.

Figure 1 Distribution map of the five sampling areas from North to South of the Valencian Community, with the approximate distance (in kilometers) in a straight line between the wetlands studied.

The Albufera of Valencia is not a sampling area; it is a resting and feeding area at the end of the reproductive season.

The Pego-Oliva Natural Park is located in northeastern Alicante Province (38°52′50″N, 0°04′09″W; Fig. 1) and comprises ca. 1,290 ha in total. Among them, the permanent wetland areas sum ca. 800 ha. In a period of 30 years (1961–1990) the mean annual precipitation is 817 mm; however, in our studied period (2002–2009) this mean was 1,014 mm. These latter years were of large water affluence, where the average in our study period exceeded by 200 mm the average of three previous decades. The mean annual temperature is ca. 17 °C; however, in our studied period this mean was 19.5 °C, with the maximum monthly mean temperatures being 30.8 °C in July and August, with the absolute maximum temperature recorded of 43 °C in July (1961–1990). The mean temperature in July was 25.1 °C (mean period 2002–2009), fitting with the species requirements of at least 20 °C mean temperature in July for breeding (Cramp, 1985). Large extensions of the wetlands are currently occupied by rice fields. The dominant plant species in the wetlands of the Valencian Community that are used by the Whiskered Tern to build their nest are Phragmites australis, Thypha dominguensis, T. latifolia, Scirpus tabernaemontani and S. maritimus, being very similar to those found in other studied areas in Iran (Barati, Aliakbari & Ghasempouri, 2011). Other species also occur in the area, such as Tamarix gallica in adjacent areas to the wetlands, and interesting aquatic vegetation in the channels and ponds, such as Myriophillum verticillatum, Potamogeton sp. or Chara sp.

The Hondo Natural Park (38°11′20″N, 0°45′12″W; 2,495 ha) had an average annual precipitation of 300 mm in the years of study, being three times less than the average in the Pego-Oliva N.P. The Xeresa marsh (39°01′17″N, 0°11′37″W; 437 ha), the Moro marsh (39°37′14″N, 0°15′34″W; 800 ha) and the Almenara marsh (39°44′53″N, 0°11′17″W; 1,497 ha; Fig. 1) all present more similar habits than the Pego-Oliva N.P. (although lacking rice fields), with suitable vegetation to build and anchor their nests and relatively stable water levels, and therefore are also chosen by the Whiskered Tern to breed.

Methods

The reproductive success (hatching, fledgling and breeding success) of the species was studied in relation to water level variation either produced by sudden climatic changes or hydrological regimens, such as continuous rain or hail, or anthropogenic causes such as marsh draining to irrigate crops or channeling works near the breeding area.

From 2003 to 2008 in Pego-Oliva N.P. we tried to control water level fluctuations and maintain it as stable as possible, using sluices in the channels that supply water from the marsh to the breeding area. However, it was not always possible since large amounts of water entered to the marsh by rain, surface runoff, and rivers flow, which exceeded the water discharge from the marsh and quickly raised water levels.

Data collection

Depending on the area and colony evolution, wetlands and colonies were visited once a week to record data, when the birds arrived to the wetlands for colony establishment (April–May). Visits were extended twice per week from egg-lay to hatching period (May–July); at least twice per week from the first egg laid (June–July) to fledgling time (a chick needs ca. 21 days to fly), and once or twice per week (depending of the colony) in the fledglings’ period (June–August). Nest monitoring was carried out using labeled wooden stakes, on which the number of the nest was marked. We tried to reduce the working time in the colony to minimize possible disturbance to the nests and nestling.

We also established a typology on the quality of the nests to explore whether the nest structure affect to the hatching success, differenciating among high, medium and low nest quality. High quality nests were considered those that formed a platform made with fragments of different plant species which were weaved together to form a solid, concave structure above water level (Fig. 2A). Medium quality nests were considered those with platforms above water level but formed with fragments of vegetation not tightly weaved and there is no evident concave central structure to support the eggs. This category usually included nests set on floating rizhomes of Phragmites sp. and even fallen Tamarix sp. branches (Fig. 2B). Finally, low quality nests were those with very low density of vegetation fragments that leave the eggs very exposed, which can roll and easily fall to the water (Figs. 2C and 2D).

Figure 2 Forms and composition of Whiskered tern nests with various qualities of structure: (A) high, (B) and (C) medium & (D) low quality.

To take census of pairs, nests, eggs and nestlings, researchers directly accessed the breeding colonies, on foot or by an inflatable boat. The number of eggs and/or nestlings present in each nest were annotated each time the colonies were visited with the aim to compare the evolution of these parameters with time. Several nestlings were ringed in the studied colonies with the capture-mark-recapture method. To reduce the underestimation, nestlings were carefully searched while approaching to the nests. In some cases, colonies were observed from the distance with binoculars where vegetation was not an obstacle. With the aim of counting also the chicks that were not in the nests at census time, researches waited for 5–20 min at a certain distance to annotate the chicks that return to the nests after entry to the colony by researchers.

Fledglings were counted when they and their parents left the nests flying after researchers approach to each nest. In some occasions, with adverse climatologic conditions, it was not possible to observe the fledglings, since the parents moved the chicks to other areas, some times more than 1 km apart from the original breeding area, and sometimes to other wetlands.

To avoid understimation of the reproductive success, a trapping technique could have been used for chicks census, as proposed by Ledwoń, Betleja & Neubauer (2015), which does not affect to the breeding success (Ledwoń, Betleja & Neubauer, 2016). However, we rejected the idea of using traps in our study to decrease the anthropic impact on the colony that could affect to the reproductive success results.

Nestlings leave nest temporarily when they are 3–10 days old and seek refuges away from it, moving further if disturbed (Cramp, 1985; Muzinić & Delić, 1997), and often return to it afterwards (Paillisson, Latraube & Reeber, 2008).

The criteria for what is considered a fledgling varies from different authors. Barati, Aliakbari & Ghasempouri (2011) consider a fledgling to a nestling unless it was found dead around the nest. Brunton (1997) considers a nestling like a fledgling at the age of 14 days old in the Sterna sp. Bollinger (1988) says that chicks of Sterna hirundo older than 10 days old have a 90% chance of becoming fledglings. However, Van de Pol et al. (2010) consider that in gull, common tern and avocet chicks younger than 14 days old, are at risk of drowning or hypothermia. For calculations on our study, fledglings were considered as those individuals older than 10–12 days old. In addition, if the chick has left the nest, their parents must remain near the original nest, and then the nest should be in good condition due to parents’ care. Moreover, we should know if the nest was active, annotating when the nests were abandoned, in relation to the structure of nests and the presence of freshly added vegetation to maintain the nest floating.

Parameters studied

During the eight years of study in the Valencian Community, 94 samplings were carried out in 14 different colonies, some of them with subcolonies (a total of 18 sampling areas). The following direct observations were made: the number of nests, the number of eggs, the number of nestlings, the number of fledglings, the date of the first fledglings, and the date when the adults with fledglings left the colony (that sometimes helped us to calculate the number of fledglings in some colonies). However, these results include all the variables that may affect these results, such as rainfall, changes in water fluctuation, predation, and nest anchoring depth.

To calculate the survival percentage of nests in the different localities along the whole study period the Mayfield method was used (Mayfield, 1961). Hatching success was calculated as the proportion of eggs that hatched successfully. Fledgling success and breeding success were calculated as the proportion of chicks that subsequently fledged successfully and the proportion of eggs that produced fledglings.

Mean values are given with ±SD. Obtained data on the studied parameters are expressed as total mean values for the eight years of study. The percentage of hatching, fledgling and breeding success were calculated for every nest.

The average percentage of loss of eggs, nests and chicks in each sampling for each year and area were also calculated.

Water level fluctuations were also recorded in the cited wetlands to study its influence on the colony and its correlation with the reproductive success, as well as the depth to which the nests were anchored. Permanent wooden stakes calibrated in centimeters were used to observe the water level fluctuations and the depth.

Analitical procedures

We wanted to compare the correlation between the average daily water fluctuation rate (measure based on the water height) and the reproductive success, and between the anchoring depth of nest and the reproductive success. For this, we performed a regression analysis relating the percentage of hatching, fledgling, and breeding success with the average daily fluctuation rate and the depth, for the selected areas and the eight years of study, among the total nests and among nests of each area. The daily fluctuation rate (cm/day) is given by the division of the water level fluctuation with respect to the sampling interval.

Data failed the test for normality (the Kolmogorov-Simirnov test), so we used the Kruskal–Wallis test to compare the dependent variables (hatching, fledgling and breeding success) among the five studied wetlands and the 18 sampling areas. We also compared the differences in the percentage of nests, eggs and chicks loss between the areas and years. All statistical analyses were performed with IBM SPSS Statistics version 22.0 (IBM, Armonk, NY, USA). Significance level was established at p < 0.05.

Results

Reproductive success

In all wetlands studied in the Valencian Community (2002–2009), a total of 663 nests, 1,618 eggs, 777 nestlings and 225 fledglings were counted.

The mean (±SD) survival percentage of nests among the different localities in the whole studied period using the Mayfield method was 61,7 ± 27%.

The mean (±SD) of the percentages of hatching obtained in high quality nests (n = 65) was 44.1 ± 43.4%; in medium quality nests (n = 120) was 38.1 ± 37.9%; and in low quality nests (n = 28) was 29.8 ± 38.3%. The total hatching success in all studied areas was: high quality (39.38%), medium quality (34.02%) and low quality (26.60%).

Considering all colonies studied, the total mean percentage of hatching success (±SD) was of 49.0 ± 25.8% (n = 663) with a total mean of clutch size (±SD) of 2.37 ± 0.27 eggs/nest. The mean of nestling (±SD) per nest was 1.1 ± 0.6. The total mean percentage of fledgling success (±SD) was 29.1 ± 21.9% with a mean of fledgling (±SD) per nest of 0.49 ± 0.4. The total mean percentage of breeding success (±SD) was 21.8 ± 18.9% (Table 1). The hatching success, as well as the fledgling and breeding success varied significantly among wetlands and among sampling areas, according to the Kruskal–Wallis test (Table 2).

Table 1 Breeding parameters of Whiskered Tern Chlidonias hybrida in the studied wetlands of Valencian Community (2002–2009) and maximum fluctuation of the water level (with mean of increasing and decreasing water level ranges).

Year	Colony No	Area	NoNest	Mean nest depth (cm) (SD)	% Hatching success (SD)	% Fledgling success (SD)	% Breeding success (SD)	Maximum fluctuationcolony (cm)	
2002	1	Pego	111	30,8 ± 0,8	72,5 ± 35,4	10,5 ± 24,3	7,7 ± 17,5	10	
2003	2	Pego	64	74,5 ± 1,5	65,3 ± 28,7	41,9 ± 38,6	27,7 ± 27	−5	
2004	3	Pego 1	79	25,9 ± 5,1	58,9 ± 38	13 ± 21,3	10 ± 16,4	14	
2004	4	Pego 2	13	62,5 ± 12,6	63,3 ± 42,2	46,7 ± 37,5	32,5 ± 25	−23	
2004	5	Pego 3	8	49,5 ± 6	77,1 ± 26,6	58,3 ± 32,1	45,8 ± 29,2	−15	
2004	6	Hondo	77	46 ± 14,2	18,9 ± 29,4	7,2 ± 23,9	4,2 ± 13,1	−30	
2004	7	Moro 1	50	35,1 ± 2,8	44,8 ± 35	19,8 ± 38,2	8,7 ± 16,5	−10	
2004	8	Moro 2	5	28,2 ± 3,9	90 ± 22,4	83,3 ± 23,6	73,3 ± 25,3	−11	
2005	9	Pego 1	18	40,9 ± 3,2	20 ± 31,6	13,3 ± 35,2	6,7 ± 17,6	−8	
2005	10	Pego 2	25	74,2 ± 6,7	61,1 ± 42,8	13,9 ± 28,7	11,1 ± 21,4	−16	
2005	11	Xeresa	114	18 ± 8,9	43,2 ± 41,8	10,8 ± 26,8	9,2 ± 23,5	−20	
2005	12	Moro	30	46,7 ± 0,6	21,8 ± 34	20,7 ± 41,2	15,5 ± 33	−1	
2006	13	Pego	27	75 ± 14,1	13 ± 26,3	1,9 ± 9,6	1,8 ± 9,6	−20	
2007	14	Pego	38	54,5 ± 7,8	21,1 ± 30,5	30 ± 45,5	14,4 ± 21,7	−6	
2008	15	Pego	10	49 ± 1,4	10 ± 31,6	10 ± 31,6	10 ± 31,6	8	
2008	16	Almenara	12	26,6 ± 2,3	80,6 ± 38,8	48,6 ± 42,9	47,2 ± 43,1	−6	
2009	17	Pego	64	25,5 ± 15,2	60,5 ± 30,2	43,4 ± 43,2	26,7 ± 26,1	6	
2009	18	Moro	32	31 ± 3,1	62,5 ± 38,2	46,7 ± 41,4	35 ± 29,6	3	
Mean (±SD)			43,2 ± 34,5	44,1 ± 18,2	49 ± 25,8	29,1 ± 21,9	21,8 ± 18,9	Rank Mean
+ 8,2 ± 4,1
−12,7 ± 8,2	

Table 2 Statistics of Kruskal–Wallis tests used to check the differences in the percentage of hatching, fledgling and breeding success according to the study areas.

The chi-square value, degrees of freedom and p-value are shown for each test.

	% Hatching success	% Fledgling success	% Breeding success	
χ2	62.405	34.440	36.577	
df	4	4	4	
P	<0.000	<0.000	<0.000	

In Pego-Oliva N.P., the area with data for the eight years of study, a total of 369 nests, 885 eggs, 493 nestlings and 136 fledglings were counted. The mean of clutch size (±SD) was 2.35 ± 0.31 eggs/nest. The total mean percentage of hatching success (±SD) was 47.5 ± 25.7% (n = 368), varying significantly among years (Kruskal–Wallis test: H (7, N = 663) = 90.24761; p < 0.0001), from 90% in years with favorable conditions to 10% in years with unfavorable conditions % (Table 1; Fig. 3). The mean of nestlings (±SD) per nest was 1.1 ± 0.6. The total mean percentage of fledgling success (±SD) was 25.7 ± 19.0%, also varying significantly among years (Kruskal–Wallis test: H (7, N = 663) = 84.53519; p < 0.0001), with a mean of fledglings (±SD) per nest of 0.43 ± 0.3. The total mean percentage of breeding success (±SD) over the eight years was 17.7 ± 13.6%, and it also varied significantly among years (Kruskal–Wallis test: H (7, N = 663) = 78.77938; p < 0.0001).

Figure 3 Hatching, fledgling and breeding success mean in Pego-Oliva N.P. (2002–2009).

Anchoring depth

The total depth average (±SD) where the nests were anchored in all sampling areas was 44.1 cm ± 18.2 cm (n = 18 colonies), with a common range between 25 and 80 cm. We recorded some nests in the Almenara marsh (2008) that were built in areas with a water depth of ca. 200 cm, being anchored on the algae Cladophora sp. attached to Myriophyllum verticillatum, but we did not obtain data from that colony.

It was considered whether a correlation exists between the water depth in which the nests are anchored over macrophytes and the hatching, fledging and breeding success, since depths strongly varied among the studied colonies. However, we did not find a correlation among the water depth and reproductive success considering all nests, showing a R2 < 0.1 (Table 3; Figs. 4A–4C). The same occurred differentiating the nests by wetlands, with R2 < 0.2. This calculation was only done for Pego-Oliva N.P. and Moro wetland, since in Hondo, Xeresa and Almenara we did not obtain enough data for a regression.

Table 3 Statistics of the linear regressions between the anchorage depth for nests (cm) and the percentage of Hatching, Fledgling and Breeding success, and between the average fluctuation rate (cm/day) and the percentage of hatching, fledgling and breeding success.

The coefficient of correlation (R), the coefficient of determination (R2) and the standard error are shown for each regression. For Hondo, Xeresa and Almenara it is not possible to do a linear regression because there was not enough data.

		Anchoring depth	Fluctuation rate	
		R	R2	Standard error	R	R2	Standard error	
	% Hatching success	0.041	0.002	40.345	0.007	0.000	40.378	
All data	% Fledgling success	0.173	0.030	33.955	0.162	0.026	34.023	
	% Breeding success	0.129	0.017	24.835	0.119	0.014	24.867	
	% Hatching success	0.112	0.013	38.619	0.163	0.027	38.344	
Pego	% Fledgling success	0.253	0.064	33.214	0.090	0.008	34.192	
	% Breeding success	0.228	0.052	23.207	0.035	0.001	23.819	
	% Hatching success	0.441	0.194	35.065	0.091	0.008	38.906	
Moro	% Fledgling success	0.233	0.054	40.925	0.097	0.009	41.886	
	% Breeding success	0.247	0.061	28.625	0.18	0.032	29.056	

Figure 4 Percentage of hatching, fledgling and breeding success according to the anchoring depth of nests (A, B & C) and the average fluctuation rate (D, E, & F) for the data set.

Daily rate of fluctuation against the percentages of nests, eggs and chicks loss

The total mean percentage (±SD) of lost eggs in the studied wetlands in the Valencian Community was 51.6 ± 25.2% with a total of n = 841 lost eggs and a mean (±SD) of 46.7 ± 47.2 eggs/colony. The total mean percentage (±SD) of lost chicks was 56.0 ± 25.7% with a total of n = 552 (Table 4) and a mean (±SD) of 30.67 ± 39.28 chicks/colony.

In Pego-Oliva N.P., the total mean percentage (±SD) was 53.3 ± 25.1% n = 392 lost eggs and a mean of 35.6 ± 27.1 lost eggs per colony. The mean percentage (±SD) of lost chicks was 57.45 ± 27.68% with a total of n = 357 a mean (±SD) of 32.45 ± 43.36 chicks/colony.

Although reproductive success does not have a homogeneous variability, it was observed (using Kruskal–Wallis test) that the reproductive success varied significantly in the interaction among the different daily rate of water level fluctuation and the different areas (Table 3).

The mean percentage of nests, eggs and chicks loss (±SD) was of 25.60 ± 21.79%, 32.06 ± 27.58% and 31.91 ± 21.28% respectively (Table 4), although some of these values varied significantly among colonies and years (Table 5). The percentage of eggs loss was different among colonies and years, and the chicks loss was different among years.

Table 4 Percentages of nest, eggs and chicks lost during the sampling period.

Eggs, chicks and fledglings in the colonies. The percentages indicate mean eggs and chicks loss, and percentages (±SD) of nest, eggs and chicks lost during the sampling period.

Colony	Year	%Meannest losssampling	%Meaneggs losssampling	%Meanchicks losssampling	N°Eggs/ colony	%Eggs loss/ colony	N°chicks/ colony	%Chicks losscolony	N°fledglins/ colony	
		X ± SD	X ± SD	X ± SD						
Pego	2002	27.67 ± 16.37	29.51 ± 39.8	76.14 ± 12.68	193	32.12	131	86.3	18	
Pego	2003	1.82 ± 2.5	24.9 ± 31.24	36.02 ± 25.97	171	36.84	108	60.2	43	
Pego 1	2004	28.72 ± 25.52	28.60 ± 31.01	49.93 ± 37.4	225	25.00	133	83.5	22	
Pego 2	2004	17.57 ± 15.79	14.94 ± 21	50 ± 44.72	23	40.89	15	46.7	8	
Pego 3	2004	7.5 ± 16.77	5.72 ± 12.79	6.2 ± 8.53	20	34.78	15	40.0	9	
Hondo	2004	37.76 ± 22.53	46.9 ± 32.49	40.45 ± 22.03	172	81.98	31	74.2	8	
Moro 1	2004	19.14 ± 15.22	28.34 ± 19.82	28.94 ± 16.42	118	55.08	53	79.2	11	
Moro 2	2004	8 ± 17.89	14.3 ± 20.22	11.1 ± 19.23	11	9.09	10	20.0	8	
Pego 1	2005	18.6 ± 38.17	30.06 ± 39.62	33.3	32	81.25	6	66.7	2	
Pego 2	2005	19.2 ± 42.93	30.82 ± 39.18	15.85 ± 2.19	38	36.84	24	83.3	4	
Xeresa	2005	24.80 ± 42.15	28.72 ± 40.04	41.63 ± 33.04	298	57.38	127	79.5	26	
Moro	2005	61.65 ± 30.62	74.55 ± 35.99	12.5 ± 17.68	59	77.97	13	30.8	9	
Pego	2006	31.1 ± 35.3	45 ± 39.69	14.3	50	86.00	7	85.7	1	
Pego	2007	50	41.2	71.4	34	79.41	7	28.6	5	
Pego	2008	90	90	0	20	90.00	2	0.0	2	
Almenara	2008	9.72 ± 15.28	13.15 ± 9.58	6.18 ± 12.01	28	21.43	22	50.0	11	
Pego	2009	0	13.6 ± 7.64	20	79	43.04	45	51.1	22	
Moro	2009	6.67 ± 11.55	16.7 ± 21.09	28.59 ± 24.76	47	40.43	28	42.9	16	
TOTAL	X±SD	25.60 ±21.79	32.06 ±27.58	31.91 ±21.28	89.9 ±85.5	51.64 ±25.2	43.2 ±47	56.0 ±25.7	12.5 ±10.6	

Table 5 Statistics of the Kruskal–Wallis tests used to check the differences in the percentage of nests, eggs and chicks loss between the colonies and between the years.

Differences between the years only for Pego-Oliva N.P. and Moro marsh areas are also tested (the only ones that data are available for different years). The chi-square value, degrees of freedom and p-value are shown for each test.

		% Nest loss	% Eggs loss	% Chicks loss	
	χ2	32.336	17.284	19.484	
Colonies	df	13	13	12	
	P	0.002	0.187	0.078	
	χ2	16.919	4.766	15.924	
Years	df	7	7	7	
	P	0.018	0.689	0.026	
	χ2	19.557	8.488	6.87	
Pego	df	7	7	6	
	P	0.007	0.292	0.333	
	χ2	4.653	3.773	0.563	
Moro	df	2	2	2	
	P	0.098	0.152	0.754	

In some years there were almost no losses of nests with values of 1.82% while in other years the loss has been almost absolute with maximums of 90%. The same patterns occurred with eggs and chicks obtaining minimums of 13.1% and 6.2% respectively and maximums of 90% and 76.1% respectively (Table 4).

Considering the studied variables, according to the linear regressions there is no correlation among the hatching, fledgling and breeding success and the average daily fluctuation rate, with R2 < 0.1 (Table 3; Figs. 4D–4F). However, a tendency was observed on absolute values of maximum water level fluctuation that exceed 8 cm that caused a decrease in the hatching and fledgling success percentages. This pattern is even more evident when the maximum water level fluctuation during the whole breeding period is more pronounced (Table 1).

Discussion

Whiskered Tern breeded in the wetlands of the Valencian community almost every year in the last few decades. Each year this species searched for the most suitable areas for breeding, coming back the following year to the same breeding area when the environmental conditions were suitable. This same pattern was observed by Burger & Shisler (1980) in a study on the laughing gulls (Larus atricilla), where the reproductive success was directly related to nest location.

Our results on the water depth range where the Whiskered Tern anchors their nests in the whole study period in the Valencian Community (between 25 and 80 cm) was similar to those observed in other studies which ranged commonly between 60 and 80 cm (Cramp, 1985), although Barati et al. (2011) also found nests in deeper waters (164 ± 30 cm), which depended on the type of aquatic vegetation. We did not find a clear correlation among the water depth and hatching, fledging and breeding success (Table 3; Figs. 4A–4C), probably due to the variety of factors that can influence the breeding, such as the kind of plants used for the nesting or the rain.

Our study, including our field observations, evidence that the quality of the nests and the quality of the cares that the parents provide to their descendents affect to the hatching success. For instance, a higher nest quality increases the reproductive success, as it was also observed by other authors in this family of birds and of others species (Nichols, Spendelow & Hines, 1990; Bakaria et al., 2002; Paillisson et al., 2006; Álvarez & Barba, 2008; Monticelli & Ramos, 2012; Ledwoń, Neubauer & Betleja, 2013).

The annual average clutch size (number of eggs per nest) obtained in the Pego-Oliva N.P. was 2.35 ± 0.31 (SD, n = 369), being similar those obtained in other studies, such as in Portugal, that showed averages of 2.91 (n = 45) in 1993 and 2.95 (n = 39) in 1994 (Catry, Tomé & Cardoso, 1997), and in Croatia 2.64 ± 0.77 (SD, n = 253) in 1993 (Muzinić & Delić, 1997). In other studies in France annual average clutch size was 2.71 ± 0.49 (SD, n = 211) in 2004; 2.05 ± 0.78 (SD, n = 406) in 2005 (Paillisson et al., 2006) and 2.35 ± 0.05 (SE, n = 207) in 2006 (Paillisson, Latraube & Reeber, 2008). Similar studies in Iran obtained averages of clutch size of 2.39 ± 0,1 (SE, n = 53) in 2005 (Barati, Aliakbari & Ghasempouri, 2011).

The mean (±SE) of chicks that become fledglings in all studied wetlands was 0.5 ± 0.1 chicks/nest per year, with n = 663 nests. These results are similar to those obtained in other studies in Iran, such as 0.66 ± 0.11 chicks, with n = 68 in 2008 (Barati et al., 2011), although this same author obtained in 2007 a mean of 1.65 ± 0.14 chicks with n = 75.

In the Pego-Oliva P.N. wetland, the mean (±SD) percentage of nests with one egg (between 2002 and 2009) was 8.7 ± 10.7%; with 2 eggs 38.8 ± 29.7%; with 3 eggs 41.6 ± 32.7%; and with four eggs 0.9 ± 1.5%. There are some studies in which the clutches with 5 eggs or more per nest are 0.3% of nests in Chlidonias hybrida, 1.8% in Larus cachinnans and 4.1% in L. melanocephalus (Betleja, Skórka & Zielińska, 2007). Muzinić & Delić (1997), in a study on C. hybrida and 105 nests, obtained percentages of 10.5% with 1 egg; 17.1% with two eggs; 60% with three eggs; 2.9% with four eggs; and 0.9% with five eggs, counting also the nests with no eggs, being 8.9% in 1993.

Our study, including all 18 colonies and the whole period studied, obtained a mean of clutching success of 49.0 ± 25.8%, n = 663 nests, being very similar to that obtained for the Pego-Oliva N.P. alone, which was of 47.5 ± 25.7%, n = 369 nests, with significant differences among colonies, years and areas, varying from 10% and 90% (Table 1). Other studies showed much higher clutching success. For example, Bakaria (2013), in his studies in Algeria, obtained a mean (±SD) of clutching success of 82.9 ± 34.35% in 2005, n = 302 nests and 76.6 ± 38.9% in 1996, n = 169 nests. Amini Nasab, Behroozi Rad & Riahi Bakhtiari (2010) in Iran obtained percentage between 57 and 83%. In Croatia, Muzinić & Delić (1997) found a mean of 88%, n = 96, and Ledwoń & Neubauer (2017) 82.9% in various studies in Poland between 2006 and 2015. In other continents, clutching success was 71% in the Volga in Russia, or 65.6% on Chlidonias hybrida javanicus in Australia (Cramp & Simmons, 1977).

Our low percentage of clutch success with regards to similar studies in other countries, as mentioned above, can be explained by the nature of the vegetation on which the Whiskered Tern anchor their nests. In our study area, nests are anchored on delicate floating vegetation, such as Myriophyllum sp., on the ground in shallow waters by accumulations of vegetation fragments above water level, or on fixed vegetation such as Phragmites sp. or Typha sp. Therefore, nests are strongly affected by water level fluctuations as they are not able to move vertically and eggs, chicks or fledglings can be damaged. On the contrary, nests in other countries are usually set on solid floating vegetation, such as Nymphoides peltata (Gwiazda & Ledwoń, 2015) or Nymphaea alba (Paillisson et al., 2006) and therefore water level fluctuations do not strongly affect to the nests as these are able to adapt water level variation by floating, and therefore clutching success increase.

The loss of eggs, nestlings and fledglings caused by predators was not considered independently in our study mainly due to technical problems in recording those data, and therefore must be assumed to be included in our total loss data, although we assume that they were low as pointed out in our statistical analyses.

Only in 2003 in the Pego-Oliva N.P., where the climatic conditions and water level fluctuations were stable, was a percentage of egg loss calculated, most probably due to predation. A total of 11 eggs were lost from n = 153 eggs, being a total of 7.2% of eggs loss. However, we do not have evidences that all eggs were predated and probably some of them felt to the water by parents activities. Studies in Croatia in 1993 with n = 253 eggs in 22 nests showed 8.7% of eggs loss probably due to predation (Muzinić & Delić, 1997).

A colony with synchrony helps against predation (Cramp, 1985; Quintana & Yorio, 1997) and some studies evidence that peripherial nests present lower reproductive success than those in the centre of the colony (Wittenbergejr & Hunt, 1985; Brown & Brown, 1987; Wiklund & Andersson, 1994; Minias, Janiszeski & Lesner, 2013), although other studies do not support this conclusion (Brunton, 1997). In our studies, depending on the year and the vegetation cover, the nests were more concentrated or dispersed, even in different areas with different depths.

Hatching, fledgling and breeding success in relation to the water level fluctuations

Although some authors have reported the abandonment of the eggs layings when water level fluctuations are severe (Catry, Tomé & Cardoso, 1997; Paillisson et al., 2006; Ledwoń, Neubauer & Betleja, 2013), in our study no significant differences were observed for hatching success in relation to water level fluctuation rate (Table 3; Figs. 4D–4F). A mean of ca. 50% hatching success has been found in most of our studied wetlands, excepting the Hondo N.P. with a 19% hatching success due to channelling works that caused the loss of many layings (Table 4). A large variability (namely, heterocedasticity) (Glass, Peckham & Sanders, 1972; Lix, Keselman & Keselman, 1996) in the water level fluctuations in all studied wetlands had also been observed.

In the same way, no significant differences have been observed in the fledgling and breeding success in relation to water level fluctuations and considering the different wetlands studied.

In general, a large variability exists in the breeding success, considering both eggs and nestlings that probably are influenced by other factors not considered in this study. From 1618 eggs counted in the 18 colonies in all years and areas in our study, 841 eggs (51.97%) were lost. This can be explained by the role of other factors that influence the hatching success, such as the direct effect of rain, hail or predation or the low quality of the parents’ care of the eggs which were not included in the statistical models. It has been also observed that the hatching success in some colonies was not affected by the water level fluctuation due to the fact that the parents are able to reconstruct the nests while progressive water level fluctuations and the eggs and or nestlings survive.

Following Bayard & Elphick (2011), the increase of the water level in marshes and breeding areas of Ammodramus caudacutus is one of the main causes of low breeding success, as it also occurs with Chlidonias hybrida.

We also observed that unfavorable climatic events, such as intense rain, hail, or wind or streams of water caused by human activities, also had the capacity to disrupt incubation or the destruction of nests, the loss of many layings and even the abandonment of the whole colony. In two occasions (in the years 2005 and 2008) in Pego-Oliva N.P, the rain with hail at the beginning of the summer destroyed many nests, and watering of crops in Xeresa marsh in 2005 completely desiccated the wetland, causing the loss of most layings. Channeling works (as occurred in Hondo N.P. in 2005) using heavy machinery also caused the decrease of the water level in some areas which affected the anchored nests. Furthermore, on some occasions, colonies were damaged by streams of water caused by the transfer of masses of water within the wetlands by human activities, although water level remained more or less constant. All these events caused the loss of high percentages of nests, eggs and nestling, forcing the parents to produce second clutches or to abandon the colony, moving to a different feeding area, like Albufera of Valencia (Fig. 1) or other wetlands.

This behaviour was also reported by Rizi (1994), Tomialojć (1994), Elkins (1996), Bakaria et al. (2002), Amini Nasab, Behroozi Rad & Riahi Bakhtiari (2010), Máñez et al. (2004), Van de Pol et al. (2010) and Ledwoń, Neubauer & Betleja (2013), being similar to the results obtained from studies on Chlidonias niger (Chapman-Mosher, 1978) where losses caused by severe precipitations were 37%.

In our study, the total percentage mean loss of nests, eggs and chicks (±SD) among the 94 sampling period was 25.60 ± 21.79%, 32.06 ± 27.58% and 31.91 ± 21.28% respectively (Table 4).

We observed that colonies with severe maximum water level fluctuation during the nesting period usually have low reproductive success (Table 1). In the colonies where this was not reflected, it was due to the fact that fluctuations did not directly damage the laying eggs in the breeding period due to the parents’ care. In years where there were major water level fluctuations and heavy rain, the percentages of hatching, fledgling and breeding success in Pego-Oliva N.P. were lower (with a breeding success very low, varying from 1.8%, in 2003 to 11.1% in 2005). In 2003, 2004 and 2009, where conditions were very favorable, the breeding success was higher, being 28.2%, 29.4% and 26.6% respectively (Table 1).

In the Fig. 5, two nests can be observed, one with three eggs and a high quality structure (Fig. 5A) and another one destroyed after the water level rise (17–22 July 2002), in which the level rose more than 7 cm (Fig. 5B). In addition, there was evidence of water run in the lagoon. Nests of medium or low quality were commonly found destroyed after severe water level fluctuations, differing from nests of high quality that in general resisted better water level fluctuations.

Figure 5 A nest of Whiskered Tern of high quality that was destroyed by the water level fluctuation.

(A) Nest with three eggs with a high quality over macrophytes; (B) nest destroyed by the rise of water level fluctuation.

We could not determine a specific value for the survival of the nests, where the loss of nests, eggs and nestlings remained stable, as it depends on each colony, event from each year and the vegetation where nests were anchored. In general, our data indicate that water level fluctuations >6 cm affected moderately to the colony and values above 10 cm, affected severely to the Whiskered Tern reproductive success. Sudden and severe water level fluctuations cause either the placement of the anchored nests above water level or being submerged in the water, hindering or making impossible nest access for nestlings or losing the eggs. These results are similar to those obtained by Van de Pol et al. (2010) who found that water level fluctuation between 10 and 20 cm in species of family Sternidae strongly affect to the chicks that are not able to return to the nest and may die due to hypothermia or drown.

Conclusions

It was not possible to establish a clear pattern among water level fluctuation and breeding success, so further studies are needed to obtain more significant results. We propose to undertake similar studies in wetlands where water levels can be regulated, with the range of depth of the anchorage of nests on the emergent vegetation being between 30 and 60 cm, which could improve the reproductive success.

As a recommendation, in water level controlled wetlands (that use sluices), for Whiskered Tern habitat management, and based on our observations, it is essential that the water level remains as stable as possible in the breeding season, which should not sharply vary more than ±6 cm in a short time period (1–2 days) once nests are anchored on the aquatic vegetation (such as: Phagmites sp., Thypha sp, Scirpus sp., Myriophyllum sp.), since this could cause in the destruction of nests and therefore the loss of eggs and nestlings although in our study it has not been possible to extract a clear pattern.

Most wetlands in the Valencian Community have sluices that serve to drain areas and irrigate adjacent crops, therefore they often have locks to distribute the water through channels. Once the date of incubation and nestling growth is known, introducing some synchrony in the nest, we could manage those risks or canalization works, etc, to be postponed to times outside the breeding season; this could minimize the impacts on the Whiskered Tern laying eggs.

Finally, we confirmed that despite having stable water levels, the spring and summer rains were causing the repeated abandonment of the clutches.

Supplemental Information

Supplemental Information 1 Water level fluctuation rate

Water level fluctuation rate in 94 samplings (2002–2009).

Click here for additional data file.

Supplemental Information 2 Breeding success

% hatching, fledgling and breeding success of all nest and colonies and years.

Click here for additional data file.

Supplemental Information 3 Pairs 1985–2015 5 areas Valencian Community

Raw data.

Click here for additional data file.

With special reference to: Mario Martínez Azorín for his unconditional support. Thanks to Maria Cristina Lorenzo for her support in the final stages of my work; to my family and friends for being always there; to Javier García Gans for contributing to ringing; to Pascual Lopez for his contribution in statistics; to Juan Jimenez for providing me with old data and who gave me permission to come in to the wetlands (Conselleria d’Infraestructures, Territori i Medi Ambient. Generalitat Valenciana); to Martin Haubeck (Seabird Journal Editor) for his first review.

Additional Information and Declarations

Competing Interests

Author Contributions

Field Study Permissions

Data Availability

The authors declare there are no competing interests.

Álvaro Ortiz Lledó conceived and designed the experiments, performed the experiments, analyzed the data, contributed reagents/materials/analysis tools, prepared figures and/or tables, authored or reviewed drafts of the paper, approved the final draft, general DATA.

Javier Vidal Mateo analyzed the data, contributed reagents/materials/analysis tools, prepared figures and/or tables, authored or reviewed drafts of the paper, approved the final draft.

Vicente Urios Moliner conceived and designed the experiments, performed the experiments, analyzed the data, authored or reviewed drafts of the paper, approved the final draft.

The following information was supplied relating to field study approvals (i.e., approving body and any reference numbers):

The Valencian Community government and managers of the Natural Parks involved in this study provided permissions to work in the cited wetlands.

The following information was supplied regarding data availability:

The raw data have been supplied as a Supplemental Dataset.

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
