# Peer review of "Reproductive success of Whiskered Tern Chlidonias hybrida in eastern Spain in relation to water level variation"

_PeerJ, doi:10.7717/peerj.4548_

## Round 0.1 · original submission · Major Revisions

Your manuscript was reviewed by three referees who all identified a number of issues with your manuscript that should be relatively easy to address in a revised version. I agree with the comments raised and believe that the referees have all been very constructive in their guidance. In particular they identified problems with the setting of the work in a broader context (and with reference to important work already in the literature); and also with respect to the lack of statistical analyses that would properly support the conclusions that you present.
I encourage you to engage very thoroughly with all of the issues that they have raised in preparing your revision, if you wish to resubmit it for consideration. Please note that one of the referees made annotations on a PDF version of your manuscript.

Reviewer 1 ·

Basic reporting

The study aims to investigate the effects of water level variation on the reproductive success of the Whiskered Tern, a declining wetland-breeding bird. The authors assessed basic reproductive parameters for five tern colonies in Spain over the period eight years and monitored losses of broods that were attributed to changes in water level produced by adverse weather conditions or anthropogenic processes (e.g. draining). Field methodology is correct and the field effort is impressive. The scientific problem is sufficiently introduced, but the discussion is very short and does not refer to similar findings in other waterbird species. English is good, but in places is needs some polishing.

Experimental design

While I feel that the study could potentially provide important information, especially for wetland management purposes, in the current form it lacks any basic statistical analyses, which could answer the questions raised by the authors. The study is purely descriptive, as it reports several cases of brood losses due to water level changes. While it is obvious that wetland drainage or heavy rains may cause losses in the broods of waterbirds, it would be much more informative to perform any kind of modelling, quantifying how water level changes affect tern reproduction. For example, the extensive data collected within this study could possibly allow to predict a border range of water level changes that are safe for the Whiskered Tern and do not cause serious losses to broods. It could also be analysed whether and how the impact of water level changes on tern reproduction is affected by habitat variables (e.g. vegetation type, wetland area etc.) or by starting water depth. Such information could be of great value for managers and would allow them to adjust water level policies to the reproductive requirements of the species.

Validity of the findings

The conclusions are not sound. Although the authors recommend that water level should not vary by more than 5 cm once the nests are established, I cannot see in the paper any analysis that would support this kind of conclusion.

·

Basic reporting

1. The manuscript needs to be carefully revised for English language errors.
2. The Discussion has lacks of few important references. Detailed remarks are provided below.
3. The manuscript needs to be substantially revised and parts rewritten before further consideration of publication. Most of my comments are suggestions to improve clarity. Detailed remarks are provided below.
4. The raw data are supplied but they are not displayed clearly.
5. There are no figures, and some data can be best represented by a figure for clarity.

Experimental design

The dataset is impressive, but there are no statistical analyses. Conclusion should be rationalized more strongly. Detailed remarks are provided below.

Validity of the findings

The question the authors seek to address is well defined. However, in the methods section there is a lot of missing information. Both field methods and the calculations of the parameters used in the study should be described much more detailed. Remarks are provided below.

Additional comments

Introduction
Line 67: Can the authors provide the raw data concerning the breeding trend of whiskered tern between 1990-2010?.
Line 70: “(table 2)” is not necessary, a reference is enough.
Line 71-72: I think that synchrony in laying exist only in small groups of nests within a colony, the whole colony is not synchronised in laying. Please rephrase this sentence.
Line 77-78: The authors state that whiskered tern does not breed in the natural wetlands of Albufera, but in their study they examined this species only on the artificial wetlands in Albufera. Please can the authors clarify this issue?
Line 80: The authors state that the main aim of the study was to examine the phenology in relation to water level variability. But in the manuscript there is no data about phenology. Please clarify.
Methods
Line 87: Please provide a map indicating the study area.
Line 89-91: There is no information about type of wetlands where the authors carried out their study. The authors should describe the studied wetlands (rice field etc.).
Line 103-109: Some of the following parameters were not used in the analysis so it is not necessary to list them: date of arrival of pairs to the colony, the number of pairs, the date of hatching, the date of the first fledglings, and the date when the adults with fledglings left the colony.
Line 106-108: I do not understand how these parameters were estimated in the field: the number of chicks/eggs in the nests, % hatching success of eggs, the % survival of fledglings and the productivity. Chicks of the whiskered tern can swim 2-3 days after hatching, so when a researcher enters the colony chicks may have left the nests, and it is impossible to find all of them among vegetation. Precisely how did the authors determine the percentage of chicks that fledged? Were the nests surrounded by mesh? Were the chicks ringed? Did the authors use capture-mark-recapture methods? How did the authors determine that chicks hatched successfully? These field methods should be described in much more detailed.

Result
Line 120-121: Please describe in the Methods section how the presented average percentages of nests were estimated? Is it the average from the mean from each year? N=8 years? Please, also provide the total (not averaged) percentages of nests which contain the respective number of eggs, a figure is an ideal way to show this result. Did the percentages of nests which contained 1, 2, 3 eggs differ between years? Please use a Chi square or similar test.
Line 124: Did hatching and fledging success differ between years? Please use a Chi square or similar test.

Table 1
“Initial Adults no.”, “Pairs no.” – are not necessary, they are not used in the Methods, Result and Discussion sections. The manuscript contain no information about how these parameters were estimated.
“Eggs/Nest no.” – how was this parameter calculated? This reviewer divided 191 by 111 and received the value of 1.7 but in Table 1 the value 2.1 (Eggs/Nests) is stated. This applies to all rows. How were the values calculated? Is it the average from all nests? If yes, please clarify and give SD or SE. I suppose that hatching and fledging success were estimated by division data from Table 1. Please clarify in the Methods how all of these parameters were estimated. Mean – please also provide the SE.

Line 160: “The percentage of nests, eggs and chicks lost in cases where populations did not suffer major environmental problems was 6.1%, 52.4%, and 49.2% respectively (Table 2, colony 2).” – why colony 2? and are these data only for colony 2?
Line 160: How exactly did the authors distinguish lost caused by “major environmental problems” and lost caused by other factors? Please clearly describe in Methods.
Line 160: Why 6.1% + 96.3% is not 100%? Please clarify
Table 2. Much too descriptive. To statistically evaluate if success was affected by water level variation or weather conditions, the authors can compare the fate of nests (dependent variable, hatched vs. not hatched) with an independent variable (for example a parameter describing water level changes a few days before the nest was lost. For successful nests the authors can use data for few days before hatching using for example General Discriminant Analysis (GDA).

Discussion
Line 169: Literature is not complete, there are many more papers about clutch size, hatching, and fledging success in whiskered tern, and also about the influence of water levels on the breeding of terns. For example:
Atamas N. S., Tomchenko O. V. 2015. THE INFLUENCE OF SPRING FLOOD WATER LEVELS ON THE DISTRIBUTION AND NUMBERS OF TERNS (ON THE EXAMPLE OF THE LOWER DESNA RIVER). Vestnik zoologii, 49: 439–446.

BARATI, A., Aliakbari, A. AND s. m. Ghasempouri. 2011. Variations in breeding success and daily nest survival of Whiskered Tern (Chlidonias hybrida) at two Iranian colonies. Russian Journal of Ecology 42: 315–320.

Betleja J., Skórka P., Zielińska M. 2007. Super-normal Clutches and Female-female Pairs in Gulls and Terns Breeding in Poland. Waterbirds 30: 629–634.

Ledwoń M., Betleja J., Neubauer G. 2015. An effective method for trapping both parents and chicks of Whiskered Terns (Chlidonias hybrida) and its impact on breeding success. Waterbirds 38: 290-295.

MINIAS, P., JANISZEWSKI, T. AND B. LESNER. Center-periphery gradients of chick survival in the colonies of Whiskered Terns Chlidonias hybrida may be explained by the variation in the maternal effects of egg size. ACTA ORNITHOLOGICA 48: 179-186.

Spina F. 1982. Contribution to the breeding biology of the Wiskered Tern in Val Campotto (Northern Italy). Avocetta 6: 23-33.

Line 188: 5 cm water fluctuations is normal for carp ponds in Poland where whiskered tern breed, there is no loss caused by this amount (5 cm) of water fluctuation. Birds abandon nests after hard weather – rain accompanied by low temperatures for several days. Because this species has floating nest, in my opinion changes in water level of 5 cm is not sufficient to destroy nests. The authors need to clarify and rationalize their conclusions. Please provide some photos of your colonies described in the Methods where (type of vegetation) whiskered tern nested.
Please clarify - nests were abandoned or they sank after hard weather conditions?.

·

Basic reporting

l. 65 – 84: Although you provide an introduction to the status of C. hybrida, you fail to place your work within the broader literature. Consider the references below for some examples of previous literature that has considered the relationship between water height and nesting success in other bird species (including other terns) in both natural and human managed systems. This is by no means an exhaustive list, but I would recommend you read into this area of the literature and discuss how your work fits in.

van de Pol, Martijn, et al. "Do changes in the frequency, magnitude and timing of extreme climatic events threaten the population viability of coastal birds?." Journal of Applied Ecology 47.4 (2010): 720-730.

Anteau, Michael J., et al. "Nest survival of piping plovers at a dynamic reservoir indicates an ecological trap for a threatened population." Oecologia170.4 (2012): 1167-1179.

Bayard, Trina S., and Chris S. Elphick. "Planning for sea-level rise: quantifying patterns of Saltmarsh Sparrow (Ammodramus caudacutus) nest flooding under current sea-level conditions." The Auk 128.2 (2011): 393-403.

Burger, Joanna, and Joseph Shisler. "Colony and nest site selection in Laughing Gulls in response to tidal flooding." Condor (1980): 251-258.

Mooij, Wolf M., et al. "Exploring the effect of drought extent and interval on the Florida snail kite: interplay between spatial and temporal scales."Ecological Modelling 149.1 (2002): 25-39.

Table 1: What happened in 2008? I see that there was a large rain event, but there was already a particularly low number of breeding pairs in this year. Were there other disturbances?

Table 2: This is a very confusing way to present your data and would be much more appropriate as a figure than a table. I think it would be much more appropriate to present in either a scatter plot (e.g. Effect of water height on % nest loss) or a bar graph (e.g. % Nest loss in years where crops were watered, years with heavy rain and years where neither occurred).

Table 2: You should split your data up to look at nest loss, egg loss and chick loss separately instead of having them all in one column.

l. 133 – 134: A basic plot visualising the variation in water heights over time might be useful.

l. 161: Is 2002 the only year with no major environmental problems? You should specify this. Also, it seems inappropriate to have information about a colony that was not impacted at all by water levels in Table 2 which is about water level variation that affected reproductive success. I think there is enough information in Table 1 to make your point.

Experimental design

l. 81: You predict that the speed at which water heights change will be important, but you don’t test this at all.

l. 104 – 105: Are “empty nests” nests that lost eggs after laying or nest sites that were never laid in?

l. 111 – 115: Is this method of using wooden stakes consistent between different years? Was the stake placed in the same place each year (or kept in the same place across the whole period of the study)? Are there potentially any other official water height measures that you can use to supplement or replace this water height methodology? If the water heights were being controlled in Pego-Oliva NP (l. 113 – 115), is there a more official water height measure at least for this site?

l. 112: You don’t show any correlation results between water level and reproductive success. This would be a very handy result to show.

l. 142 – 143: The wording here is confusing. Do terns ever rebuild nests? If so, how did you take this into account in your data (e.g. were colonies that re-nested counted as new colonies or the same colony?).

l. 143 – 144: As you are interested in the effect of water height on reproductive success, it’s important that you differentiate between damage caused just by rain/hail and that caused by increased water levels.

l. 147 – 149: Were the nests destroyed or just abandoned when water levels declined? How were they destroyed?

l. 152 – 154: If you analyse your water level data statistically, you might need to consider removing these events where a stream of water destroyed nests without a change in water height.

Results: Although you frame your study to be about the impact of water heights on reproductive success you haven’t conducted any analyses to actually look at the relationship between these variables. It seems your dataset would be ideally suited to carry out some sort of simple statistics to look at this link. For example, you could look at the mean water height during the breeding season and some measure of reproductive success (e.g. absolute or percentage fledgling production). You haven’t provided the raw water height data in the supplementary material so I can’t give a worked example.

Validity of the findings

l. 123 - 125: It’s probably more appropriate to provide the mean annual hatching success (58.3%) and fledging success (36.5%) than the total success.

l. 189: It’s not clear, either here or in the abstract, why you suggest a cap of ±5cm. You refer to Table 2 here, but I don’t see how this table provides you with this information. Perhaps with a more detailed analysis you might be able to suggest an ideal water height with more evidence. Even visualising your data outside of tables might help. Until then, this value of 5cm comes across is very speculative.

Supplementary data: As stated above, the raw water level data is not provided. This is necessary data to replicate your results in Table 2.

Additional comments

I feel the subject of the article is interesting and topical and the authors seem to have a nice dataset to study the topic. However, the manuscript fails to really investigate the relationship between water heights and nesting success in C. hybrida in any meaningful way. The long-term dataset provides an opportunity for some simple statistics to help describe the relationship between water heights and nesting success, but this is not done. Furthermore, with all results presented in tables it is hard to visualise the data. At its core this manuscript could provide a useful addition to the literature, both generally and on whiskered terns; however, major revisions would be required to reach a publishable level.

I also think there are a few areas where the manuscript would benefit from clearer English. I understand that the authors are not primary English speakers. I've provided some notes in the annotated pdf to highlight those areas that I feel need rewording.

You state in your methods that you mist net the terns and catch the chicks. You should provide details of your ethics approval for this work.

---

## Round 0.2 · Major Revisions

I acknowledge that you have undertaken a fairly major revision of your paper after the comments from the initial round of review. Your initial paper did not contain any statistics at all, and of course it is good that you have now attempted to include some analysis of your data. Unfortunately however, the analysis that you have done is very inadequate in it's present form and is certainly not acceptable. The reviewers have made a number of specific comments about some of the problems with the methods that you have used and the need to use tests that better fit your data. Hopefully those will help in a reworking of the data analysis.

In addition it is really important that you properly explain what you are testing in your methods and how you are deriving particular variables such as intensity of water fluctuations. At the moment many of the things reported in your figures and tables just don't make sense. As an example, Fig 5, purports to illustrate a correlation, and yet there are polynomial fitted curves drawn on this figure with no explanation as to what they are, how they were fitted, or what the error bars are.
Furthermore all results should be reported in the 'Results' section. As it is, most of them are in the 'Discussion' which is completely inappropriate.

I realise that you have already included an additional author on to assist you with these statistical analyses, and hopefully that author will be able to provide further assistance in revising your analyses.
Finally, whilst appreciating that English is probably not the first language of any of the authors, I'm afraid that the standard of English is still quite poor and needs further attention.

I have offered the option of further 'major revision' because the reviewers and myself feel that the data presented are potentially of interest and worthy of publication. However, although PeerJ certainly provides a useful route to publishing work of low impact, good standards of language, analysis and writing are very important. You may find that your work will be easier to publish in a local ornithological journal.

·

Basic reporting

I include remarks in "General Comments for the Author"

Experimental design

I include remarks in "General Comments for the Author"

Validity of the findings

I include remarks in "General Comments for the Author"

Additional comments

Authors have done a great job. I have only minor remarks.
Line 78-83-: What was the cause of the decline of the population? Habitat destruction?
Line 86: Please add Ledwoń et al. 2014 (J Ornithol 2014, 155:459–470)
Line 132-138: Not necessary, I suggest to delete.
Line 142-145: Not necessary.
Line 192: How exactly fledgling success was calculated? Please write it more clearly. For example “We calculated fledgling success as the proportion of chicks that hatched in relation to the number of fledgling” Please rephrase.

Line 193-194: “Multiplying hatching success”? Did you multiple percentages? Why not proportion of eggs in relation to the number of fledgling? Please rephrase.

Line 199-201: Why 11 events? How did you select these events? Please move here the reasons from line 243-244.

Line 244-245: why three means, how it was exactly calculated? Is it means for nests, eggs and nestlings? I am sorry but I do not understand. You write about 11 events from 8 colonies. I also do not understand means from lines 246-247. Please rephrase.

Line 220-225: How these categories correspond with Fig. 5?

Line 235: n=18 – what is it? Colonies? Colonies/years? Please explain
Line 265-278 – Why do you calculate breeding parameters separately for Pego-Olia N.P and all wetlands in the Valencian Community?
Line 292: Fig 4 do not show water level fluctuations.
Line 295-306: Please include, compare and discuss breeding parameters from other studies (suggested papers) with your data. You noted very low hatching (breeding, fledgling) success, please discuss the reasons in comparison with suggested papers. In Poland on carp ponds hatching success in usually about 80-90%.
Suggested papers:
Betleja J., Skórka P., Zielińska M. 2007. Super-normal Clutches and Female-female Pairs in Gulls and Terns Breeding in Poland. Waterbirds 30: 629–634.
Ledwoń M., Betleja J., Neubauer G. 2015. An effective method for trapping both parents and chicks of Whiskered Terns (Chlidonias hybrida) and its impact on breeding success. Waterbirds 38: 290-295.

Mateusz Ledwoń, Jacek Betleja & Grzegorz Neubauer (2016): Different
trapping schemes and variable disturbance intensity do not affect hatching success of
Whiskered Terns Chlidonias hybrida, Bird Study, DOI: 10.1080/00063657.2015.1136263
MINIAS, P., JANISZEWSKI, T. AND B. LESNER. Center-periphery gradients of chick survival in the colonies of Whiskered Terns Chlidonias hybrida may be explained by the variation in the maternal effects of egg size. ACTA ORNITHOLOGICA 48: 179-186.
Table 2. Please give the number of nests (n)
Fig. 1. What does the “km” on the map mean?
Fig. 3. I do not see “depth” on figure 3 , as is mentioned by Authors.
All the best,
Mateusz Ledwoń

·

Basic reporting

- I still feel that the work would benefit if the manuscript was put in a much broader context (i.e. not just about terns). For example, you could refer to nest flooding studies on saltmarsh sparrows (e.g., Bayard and Elphick 2011), laughing gulls (e.g., Burger and Shisler 1980) or a large scale study on multiple shorebirds (van de Pol et al. 2010). See below for full references. By putting your work in this broader context you could greatly improve the impact of your work, potentially outside of just those people interested in whiskered terns.
- Line 132 – 138: I think it’s only relevant to discuss the fauna of the area if it is the predator or prey of the terns.
- There is a large amount of data that you present in your results that you don’t provide any figures for. Most importantly, your ANOVA analysis, but also your general reproductive results (Lines 265 – 280).
- Make sure that you reference to your figures in order. For example, figure 2 should be the second figure you refer to in the text. At the moment, this is not consistent.
- Fig. 4: x-axis is not clear. I think it would be better to use something like ‘water level change (cm)’.
- Fig. 5: does not seem necessary. I would remove it.
- Fig. 7: This is more appropriate for an introduction.
- Table 1: You need to provide a more informative title for the table. The title you provide is specific R-code and may not be informative for all readers. Provide a title that gives a better idea of what was actually tested.
- Table 1: You should also present the residual degrees of freedom in your ANOVA table so we have an idea of the amount of statistical power you have for your test.
- It would also be good to include the ANOVA tables of the Fledgling and Hatching success analysis, or at least refer to the supplementary figures in the text (e.g. line 254).
- I think there could still be some improvements to the written English. It is not a huge issue (it is all mostly easy to understand), but it could help you convey your points more clearly. However, I feel the paper still has to go through some major changes, so I'll make more comments on this when it comes back for further review.

Bayard, Trina S., and Chris S. Elphick. "Planning for sea-level rise: quantifying patterns of Saltmarsh Sparrow (Ammodramus caudacutus) nest flooding under current sea-level conditions." The Auk 128.2 (2011): 393-403.

Burger, Joanna, and Joseph Shisler. "Colony and nest site selection in Laughing Gulls in response to tidal flooding." Condor (1980): 251-258.

van De Pol, Martijn, et al. "Do changes in the frequency, magnitude and timing of extreme climatic events threaten the population viability of coastal birds?." Journal of Applied Ecology 47.4 (2010): 720-730.

Experimental design

You have taken into account the general comments of the reviewers to include statistical analysis; however, your statistics are quite badly flawed. This is a serious issue that needs fixing. I’ve included separate comments on the stats first and then more general comments on the methods section.

Statistics
- Line 199 – 201: I think the aim of your statistics should be to look at the relationship between water heights/water height change and hatching/fledgling/breeding success. For this to have any meaning, we also need to know what success rates are like when there is no change in water height. Therefore, you should consider ALL your data, not just the data when an event occurred (if you have it).
- Line 217 – 218: You said you conducted a quadratic transformation on your hatching data, but in the supplementary material it is a square root transformation. Which one did you use?
- Line 220 – 225: I don’t understand why you have categorised your water height data, why can’t you treat it as a continuous variable as you do in Figure 4? Additionally, the way you have categorised your data is badly flawed. You are testing the effect of water height change on success, but you also based your groupings on nest success (e.g. Light events were 0-6 cm OR events that didn’t cause high nest loss). With this definition you will definitely find an effect of water height change because you have adjusted your groupings based on their success!! If you feel that you need to include some measure of the duration or speed of water change, perhaps you should make your predictor variable rate of change instead (e.g., cm/day). The way you currently structure you data is unusable and needs a serious overhaul.
- Line 226 – 229: Kruskal-Wallis is a test that we employ if the assumptions of normality aren’t met within our data. If this is the case, you need to say so. If it is not, you can simply carry out the analysis with an ANOVA. Also, you can do Kruskal-Wallis in R, so it might be easier to use all the same analysis software.
- Line 239 – 242: If you want to discuss this result you should analyse it statistically. You can do this easily with a linear model in R (e.g. lm(Success ~ Depth)).
- Line 310 - 311: You didn’t mention anywhere how you accounted for predation in your statistical analyses.
- Line 375 – 377: I guess you mean there is a large variability in how breeding success changes with water levels? The standard deviation of breeding success shouldn’t be more than the mean, otherwise there could be negative breeding success.

General comments
- Line 120 - 121: Why haven’t you presented mean annual precipitation for the full time series (i.e. 1961 – 2015)? You must have the data if you are able to determine the rainfall during your study period.
- Line 180: You only include fledglings that remain by the original nest. What about chicks those fledge but leave the area, why are these not considered?
- Line 185: Previous studies consider all chicks that weren’t found dead as fledged. If you are going to mention this you need to point out why you didn’t use this method (i.e. why is this assumption flawed?).
- Line 217 and Line 358: If the assumptions of the test were not met, this is called heteroscedastic (i.e. there is a relationship between the variability in the data and the response, in this case water height).
- You need to point out that your predictor variable, water fluctuation, is a relative measure based on the water height in that year. You currently don’t say this anywhere.

Validity of the findings

I don’t feel that I can comment on the validity of the findings until the statistical analysis has been adjusted (see above).

Additional comments

I think you have taken some steps to improving the manuscript since my last review. A major change from the previous work is your inclusion of statistical analysis (ANOVA) and more in depth figures; however your analysis structure has a number of major flaws (documented in more detail in the comments). I also think that your choice of figures could be improved. I still feel like this work can become a good paper, but very major revisions are still required. Make sure you think more carefully about how you will analyse your data. If in doubt, it may be worthwhile getting some professional statistical advice.

---

## Round 0.3 · accepted · Accept

You have done a good job revising your paper, and I believe that it is now acceptable for publication. Thank you for engaging so well with the reviewers comments.